# Nucleic Acid Testing of SARS-CoV-2

**DOI:** 10.3390/ijms22116150

**Published:** 2021-06-07

**Authors:** Hee Min Yoo, Il-Hwan Kim, Seil Kim

**Affiliations:** 1Microbiological Analysis Team, Biometrology Group, Korea Research Institute of Standards and Science (KRISS), Daejeon 34113, Korea; hmy@kriss.re.kr (H.M.Y.); ihkim@kriss.re.kr (I.-H.K.); 2Department of Bio-Analytical Science, University of Science & Technology (UST), Daejeon 34113, Korea; 3Convergent Research Center for Emerging Virus Infection, Korea Research Institute of Chemical Technology, Daejeon 34114, Korea

**Keywords:** SARS-CoV-2, PCR, isothermal amplification, genome sequencing, nucleic acid testing, reference materials

## Abstract

The coronavirus disease 2019 (COVID-19) has caused a large global outbreak. It is accordingly important to develop accurate and rapid diagnostic methods. The polymerase chain reaction (PCR)-based method including reverse transcription-polymerase chain reaction (RT-PCR) is the most widely used assay for the detection of SARS-CoV-2 RNA. Along with the RT-PCR method, digital PCR has emerged as a powerful tool to quantify nucleic acid of the virus with high accuracy and sensitivity. Non-PCR based techniques such as reverse transcription loop-mediated isothermal amplification (RT-LAMP) and reverse transcription recombinase polymerase amplification (RT-RPA) are considered to be rapid and simple nucleic acid detection methods and were reviewed in this paper. Non-conventional molecular diagnostic methods including next-generation sequencing (NGS), CRISPR-based assays and nanotechnology are improving the accuracy and sensitivity of COVID-19 diagnosis. In this review, we also focus on standardization of SARS-CoV-2 nucleic acid testing and the activity of the National Metrology Institutes (NMIs) and highlight resources such as reference materials (RM) that provide the values of specified properties. Finally, we summarize the useful resources for convenient COVID-19 molecular diagnostics.

## 1. Background

Severe acute respiratory syndrome coronavirus-2 (SARS-CoV-2) is a novel coronavirus that first appeared in Wuhan, Hubei Province, China in December 2019, connected to a seafood market [1,2]. Seven coronaviruses have been reported to infect humans; four of them, human CoV-NL63 [3], HCoV-OC43 [4,5,6], HCoV-229E [7,8], and HCoV-HKU [9,10], cause mild and seasonal respiratory tract disease, whereas SARS-CoV [11,12,13,14,15], MERS-CoV [16,17,18,19], and SARS-CoV-2 can cause severe symptoms. In particular, SARS- CoV-2 is suited to human-to-human transmission and spreads rapidly to other locations, causing lung injury, multiorgan failure, and death [20,21]. As of this date, the number of confirmed cases is still increasing, as is the number of deaths [22,23]. Therefore, an understanding of the SARS-CoV-2 host and pathogen biology is important to offer valuable insights into the diagnosis and treatment of the disease including the development of new therapies [24,25]. Here, we review the basic biology of SARS-CoV-2 including the origin, pathophysiology, and diagnosis methods.

### 1.1. Nomenclature of SARS-CoV-2

Currently, almost a million sequences of the SAR-CoV-2 genome are publicly available via the Global Initiative on Sharing All Influenza Data (GISAID) and GenBank [26,27]. Based on these genome sequences, the phylogenetic classification of SARS-CoV-2 was performed and the nomenclature of GISAID, Phylogenetic Assignment of Named Global Outbreak LINeages (PANGO lineage), and Nextstrain are widely used in scientific and clinical communities [26,28,29] The major lineages of each nomenclature are summarized in Table 1.

GISAID introduced the nomenclature system of SARS-CoV-2 based on marker mutations and named the clade with actual letters of marker mutations [30]. For example, the clade G has a characteristic mutation in the spike protein gene, D614G. In the nomenclature of GISAID, the initial strains of SARS-CoV-2 were grouped as S and L clades and the current strains of SARS-CoV-2 were classified as eight major clades (S, L, V, G, GH, GR, GV, and GRY) [30,31]. The L clade contains the reference strain WIV-04 and was the dominant lineage in early 2020. The L clade later diverged into clades V and G, and clade G diverged into clades GH, GR, GV, and GRY.

The PANGO nomenclature systems focused on the active virus lineage [29]. This nomenclature is dynamic and the lineages of the PANGO nomenclature are marked as three statuses: active, unobserved, or inactive. The lineages documented within a month are marked as active. The lineages documented within three months are marked as unobserved, and the lineages that were not documented for more than three months are regarded as inactive. The lineages of PANGO nomenclature are named with a letter and numerical values. The initial lineages are denoted as lineages A and B. Although clade B includes the first genome sequenced strain, the phylogenetic analysis suggested that the most recent common ancestor of SARS-CoV-2 was close to early lineage A [29]. The descendent lineages from initial lineages were assigned with numerical labels. The descendent lineages can be designated with the phylogenetic evidence that the descendent emerged from parental lineages and the descendants showed significant transmission to geographically distinct populations. The designated descendent lineages can also be parental of new emerging lineages and these new lineages have been labeled as parental lineages with additional numerical values. For example, a new emerging lineage from lineage B1 can be labeled as B1.1. The lineages can have a maximum of three sublevels and newly designated lineages emerging from a lineage with three sublevels will be labeled with new alphabetical letters. For example, the parental lineage of lineage C.1 is the lineage B.1.1.1.

The clades of Nextstrain nomenclature were initially named according to year–letter combinations [32]. Major clades were designated as the clade reached more than 20% of global frequency for more than two months. Based on this criteria, the initial clades were designated as 19A, 19B, 20A, 20B, and 20C. However, due to the global travel restriction, no more clades were designated according to the criteria. For this reason, Nextstrain updated their major clade designation criteria with regional frequency (>30%) and recognized variants of concern. Currently, 12 major clades (19A, 19B, 20A, 20B, 20C, 20D, 20E, 20F, 20G, 20H/501Y.V2, 20I/501Y.V1, 20J/501Y.V3) are designated in the nomenclature of Nextstrain.

### 1.2. Notable Variants of SARS-CoV-2

The first strain of SARS-CoV-2 was discovered in Wuhan, China and designated as Wuhan-Hu-1 or WIV-04 [1,33]. The comparison of whole genome sequences showed that the strain was closest to the SARS-like coronavirus RATG13 found in bats (*Rhinolophus affinis*) in China [2]. The overall genomic sequence similarity of RATG13 to SARS-CoV-2 was 96.1%. However, the spike protein gene of RATG13 lacked the furin cleavage site that is essential for the cell entry of SARS-CoV-2, indicating that RATG13 was not the immediate ancestor of SARS-CoV-2 [34,35]. After discovery of the first SARS-CoV-2 strains, SARS-CoV-2 like viruses were found in pangolins and bats [36,37,38]. The genome sequences of pangolin-derived CoVs also showed high similarity to those of SARS-CoV-2, but the furin cleavage site was missing in the spike gene sequences of pangolin-derived CoVs [35,37,38]. A bat-derived CoV, RmYN02, was identified and the genome of the virus showed high similarity to that of SARS-CoV-2 [36]. Although the sequence similarity of RmYN02 was slightly lower than those of RATG13 and pangolin-derived CoVs, the furin cleavage site was inserted, indicating that the addition of the cleavage site can occur naturally [35,36].

The D614G variants had a change in spike gene and replaced the initial strains of SARS-CoV-2 [39]. The studies on the variant D614G showed that the infectivity of the variant was increased without increased severity [39,40,41]. The engineered variants containing the D614G substitution showed more efficient infection in human cells and animal models without altering antibody neutralization and pathogenicity [41]. A population genetic analysis of COVID-19 also showed that the transmissibility of the variant was increased but there was no sign of increased mortality or clinical severity of the variants [40].

As new variants with increased pathogenicity, reduced neutralization, and/or increased transmissibility emerged, the U.S. Centers for Disease Control and Prevention (CDC) and Public Health England (PHE) classified some notable variants according to the attributes of the variants [42,43]. The CDC classified the variants according to the evidence and significance of the variants into Variant of Interest, Variant of Concern, and Variant of High Consequences. PHE classified variants as Variant Under Investigation (VUI) and Variants Of Concern (VOC). When the variants are considered to have concerning characteristics, they are designated as VUI. After a risk assessment of VUI is conducted, they can be re-designated as VOC. These notable variants are summarized in Table 2.

VOC-20DEC-01, also known as 20I/501Y.V1 or B.1.1.7, was first discovered in the United Kingdom in December 2020 [42] and is defined by 13 mutations [42]. Recent studies have estimated that the transmissibility of VOC-20DEC-01 is increased by 43–90% and a similar transmission increase was observed globally [44]. VOC-20DEC-01 was also detected in domestic cats and dogs, raising concern over human-to-animal transmission or vice versa [45]. Previously reported animal infections were asymptomatic to mild symptomatic, but VOC-20DEC-01 infection in animals showed relatively severe symptoms such as myocarditis [45]. In February 2021, the variants with a spike gene E484K mutation were reported and designated as VOC-202102/02 [42]. Another variant with N501Y mutation is VOC-20DEC-02 (20H/501Y.V2, or B.1.351), which was first discovered in South Africa. VOC-20DEC-02 is defined by 17 mutations including the E484K mutation, K417N mutation, and two deletions. The variant also showed increased transmissibility (approximately 50%) compared to previous variants [44]. The third variant with N501Y is VOC-21JAN-02 (P.1 or 20J/501Y.V3), discovered in Brazil [42,46]. The genome of VOC-21JAN-02 is defined with 17 non-synonymous mutations, four synonymous mutations, three deletions, and four insertions [42]. VOC-202101/02 almost fully replaced its parental variant within two months, indicating increased transmissibility of VOC-21JAN-02 [47,48]. Molecular clock analysis showed that the variants emerged in mid-November 2020 at which time hospitalizations rapidly increased [49].

The characteristic mutations (N501Y, E484K, and K417N) of the variants with N501Y are mutations in binding sites to viral receptor ACE2 and were already a concern prior to the discovery of these variants [50,51,52,53]. The studies on these variants showed that they had impacts on neutralization by immunity [54,55,56,57,58,59,60]. However, recent research showed that the residual immunity still provided protection, although variants reduced the efficacy of the vaccine [61].

There were also emerging variants without N501Y, E484K, and/or K417N. The characteristic mutations of A.23.1 are F157L, V367F, Q613H, and P681R [62]. A.23.1 with E484K was designated as VUI-21FEB-01 in the United Kingdom. These strains were first identified in Uganda and are spreading. One of the characteristic mutations, Q613H, is regarded as functionally equivalent to the D614G mutation of ‘G’ clade strains. B.1.427 and B.1.429 were first discovered and designated as Variants of Concern in the United States [43]. The characteristic mutations of both lineages are L452R and D614G; these variants showed increased transmissibility and reduced neutralization by convalescent and post-vaccination sera [63].

B.1.617 was the emerging lineage in India and also designated as VUI-21APR-01 in the United Kingdom [42]. B.1.167 has two characteristic mutations of different lineages: L452R and E484Q [64]. The variants were neutralized with convalescent sera of COVID-19 patients and vaccine of BBV152, although the efficacy was low [64].

## 2. PCR-Based SARS-CoV-2 Detection

### 2.1. Reverse Transcription Quantitative PCR (RT-qPCR) Method

Detection of the SARS-CoV-2 viral genome, consisting of single-stranded RNA, is effectively done by reverse transcription quantitative polymerase chain reaction (RT-qPCR), which is the gold standard technique widely used in molecular diagnostics [65,66]. There are several practical considerations when performing diagnostic assays using RT-qPCR.

(1) Sample quality: RT-qPCR tests are presently being used for the identification of SARS-CoV-2 in clinical specimens such as upper respiratory tract specimens (saliva, oropharyngeal swab-OPS, nasopharyngeal swab-NPS, nasal swabs), lower respiratory specimens (sputum, bronchoalveolar lavage-BAL, endotracheal aspirate-ET, fibrobronchoscope brush biopsy-FBB), blood (serum, plasma), urine, feces, rectal/anal swabs, stool, and corneal secretion [67,68]. To check the sample quality of clinical specimens from different origins, an RNA isolation procedure is required to obtain purified high-quality RNA from the samples, which then needs to be analyzed using chip-based capillary electrophoresis (such as the Agilent Bioanalyzer system), electrophoretic separation on a high-resolution agarose gel, and spectrophotometry [69].

(2) Reference curve: Data processing can critically affect the analysis of RT-qPCR results [70]. PCR data processing is based on standard curves or on PCR efficiency assessments [70]. Standard curves are used to assess RT-qPCR efficiency without theoretical and practical problems [70]. The estimation of RT-qPCR efficiency using standard curves usually involves the serial dilution of a concentrated stock solution, after which standard samples are analyzed through RT-qPCR by measuring the quantification cycle (Cq) using standard procedures [70]. The most widely used Cq value is the threshold cycle (Ct), the cycle at which the expression of a target gene first exceeds a calculated fluorescence threshold level [71]. For example, to detect low amounts of SARS-CoV-2 RNA, a series of diluted RNA templates are used to determine the Ct value, which can provide a standard curve for evaluating the reaction efficiency [72]. However, the Ct value itself cannot be directly explained as viral load without a standard curve using reference materials [73]. When interpreting the results of SARS-CoV-2 RT-qPCR, the validity of the standard curve should be proved using reference materials with accurate viral copy numbers to interpret Ct values as viral loads [73].

(3) Viral load: The success of virus isolation depends on the viral load [74]. Viral loads in sputum samples and throat swabs are high when obtained within seven days after initial symptoms are observed, ranging from 10^4^ to 10^7^ copies per mL. This pattern is broken as low quantity of virus are obtained from samples taken after day 8 [75]. In general, sputum samples show higher viral loads than throat swab samples, whereas low viral RNA is detected in urine or stool samples [75]. The two main factors that influence the quantitative measurement of viral roads are Cq values that are repeatable with acceptable uncertainty and a reliable means of converting from the Ct value to viral load [76,77,78]. For molecular diagnostic assays, a limit of detection (LoD) and a limit of quantification (LoQ) are also considered the lowest concentrations of target RNA that can be detected by RT-qPCR [79].

(4) Sampling methods: RT-qPCR tests for SARS-CoV-2 have shown a high variation of false-negative rates (FNR) and false-positive rates (FPR) [80,81]. Numerous methods have been developed with the goal of improving the sensitivity, safety, and rapidity of COVID-19 tests by RT-qPCR. For example, one group tested the efficiency and sensitivity of SARS-CoV-2 detection of clinical specimens collected directly in nucleic acid stabilization and lysis buffer (NSLB), a mixture of lysis buffer and RNA preservative, instead of a viral transport medium (VTM), thus inactivating the virus immediately after sampling [82].

(5) Sample source: To improve the expandability of SARS-CoV-2 testing, several sampling approaches have been developed including nasal, pooled nasal, and throat (oropharyngeal) swabs as well as saliva. Different clinical sampling methods affect the diagnostic performance of SARS-CoV-2 infection tests by RT-qPCR including sensitivity and specificity, and thus should be carefully considered [83,84,85,86]. The combined swab is largely recommended as a more appropriate specimen for diagnosis by RT-qPCR [87,88,89].

(6) Sensitivity: The conserved regions, ORF 1ab (RNA-dependent RNA polymerase, RdRp), envelope (E), and nucleocapsid (N) genes of SARS-CoV-2, are usually selected as the standard target genes for primer and probe design [90,91]. However, initial reports of SARS-CoV-2 and other coronavirus sequences gave rise to an incorrect degenerate base that did not align with the SARS-CoV-2 RNA sequence found, and there were reports regarding the decreased sensitivity of using RdRp as a target gene for RT-qPCR assays [90,92]. As the pandemic continues, many laboratories around the world rely on routine diagnostic primers and probes. Thus, proper assays can increase the sensitivity of SARS-CoV-2 detection and help prevent the further spread of the virus [92,93,94,95].

(7) Pooling technologies: The pooling of multiple swab samples before RNA isolation and RT-qPCR analysis has been proposed as a promising solution to reduce costs and time as well as elevate the throughput of SARS-CoV-2 tests for large-scale testing as in the case of schools [96,97,98,99]. For example, batch testing of over 100,000 hospital-collected nasopharyngeal swab samples from patients alleviated three quarters of testing reactions with a minor reduction in sensitivity, indicating the effectiveness of the pooling approach in the field [100,101]. Current studies suggest that the pooling of individual samples before testing should be considered to increase the reliability of SARS-CoV-2 testing throughput.

Once all practical considerations have been evaluated, there are two ways that RT-qPCR can be performed. The two-step RT-qPCR method is required to convert RNA into complementary DNA (cDNA) [102]. On the other hand, the one-step RT-qPCR method combines reverse transcription and PCR in a single tube and uses reverse transcriptase as well as a DNA polymerase [103]. The schematic procedure of RT-qPCR is shown in Figure 1A.

A critical need for rapid and accurate diagnostic methods has emerged in the clinic and public health organizations. Several PCR-based assays have been developed and are currently being used in clinical, research, and public health laboratories [104,105,106]. However, it is not clear which PCR condition they should adopt or whether the data are comparable. In response to the growing need and the lack of publicly available information, several research groups have optimized real-time PCR-based primer sets, protocols, and PCR conditions [107,108].

Independent evaluations of the designed primer–probe sets used in SARS-CoV-2 RT–qPCR detection assays are necessary to compare and select appropriate assays [90,109]. Additionally, several studies have utilized serum and stool specimens for the RT-qPCR-based detection method [110,111,112,113].

### 2.2. Reverse Transcription Digital PCR (RT-dPCR) Method

In recent years, we have seen the advance of digital PCR (dPCR) as a complementary approach for measuring nucleic acids, a technique that is highly accurate and reproducible when targeting the viral genes of SARS-CoV-2 [114,115,116,117]. The advantages of digital PCR compared to quantitative PCR include quantification without the need for calibration curves, higher accuracy, and sensitivity that may arise from sub-optimal amplification efficacy because dPCR can detect low amounts of nucleic acid [118,119]. The schematic procedure of dPCR is shown in Figure 1B.

Reverse transcriptase quantitative PCR (RT-qPCR) and digital PCR (dPCR) have been widely used for quantitative analyses of clinical samples. Recently, many groups have developed a reverse transcription droplet digital PCR (RT-ddPCR) assay for sensitive detection of the SARS-CoV-2 virus [120,121,122,123]. Optimization of the primer. and probe assays is necessary to remove false negatives or positives for clinical diagnosis of viral infection [72,124]. Multiple molecular diagnostic kits have been developed and validated for use nationwide [125]. However, the analytical sensitivity and the relative sensitivity of different kits to detect low copy number of SARS-CoV-2 viral RNA are variable [126,127].

## 3. Isothermal Nucleic Acid Amplification Methods

Although the RT-qPCR method is considered the ‘gold standard’ for SARS-CoV-2 detection [128], its limitations have stimulated the development of simple, rapid yet sensitive nucleic acid detection methods [129]. As a result, isothermal nucleic acid amplification has emerged as an alternative detection method for SARS-CoV-2 viral RNA from clinical samples [130]. In general, isothermal amplification techniques increase the analytical signal by increasing the target nucleic acid concentration through enzymatic activities at a fixed temperature, and simultaneously detecting the signal with colorimetric or fluorescence indicators [131]. Changes in color, fluorescence level, or turbidity indicate the presence of SARS-CoV-2 RNA or DNA [131]. Therefore, unlike RT-qPCR, isothermal amplification methods do not require thermal cycling instruments or specialized technicians for disease diagnosis [132]. Current isothermal nucleic acid amplification methods used for SARS-CoV-2 detection include, but are not limited to, loop-mediated isothermal amplification (LAMP), recombinase polymerase amplification (RPA), nucleic acid sequence-based amplification (NASBA), strand-displacement amplification (SDA), and rolling circle amplification (RCA) [133,134,135,136,137] (Figure 1C). Herein, we describe the general procedures and components of isothermal amplification methods commonly used for diagnosis of SARS-CoV-2.

### 3.1. Loop-Mediated Isothermal Amplification (LAMP)

The loop-mediated isothermal amplification method, coupled with reverse transcription (RT-LAMP), is the most widely used isothermal amplification technique for SARS-CoV-2 nucleic acid detection. First described by Notomi et al. [138], this method uses strand displacement activity of DNA polymerase and a set of inner and outer primers (four or six specific primer sequences) to amplify the target nucleic acids. LAMP is carried out at a single temperature between 60 and 65 °C, and generates up to 10^9^ copies of DNA in less than an hour [133,137,139]. The LAMP procedure is initiated by hybridization of the forward inner primer (FIP) toward the target DNA template, which synthesizes the complementary strand. Then, the outer primer hybridizes to the target DNA, which initiates DNA synthesis by strand displacement. Subsequently, a FIP-hybridized complementary strand is released and forms a loop structure at one end of the sequence. The corresponding sequence becomes the template for the backward inner primer (BIP), which initiates another DNA synthesis by strand displacement, and then produces a ‘dumb-bell’ like DNA structure. Self-primed DNA synthesis of the corresponding sequence then converts the ‘dumb-bell’ like structure into a ‘stem-loop’ like DNA structure. Corresponding stem-loop DNA then becomes the template for LAMP cycling, and the target DNA sequence exponentially amplifies until the reaction is completed [138]. Amplified products are detected by changes of color as the accumulation of DNA changes, pH levels, or by changes in turbidity as magnesium pyrophosphate level increases [140,141,142]. Amplified products are also detected by Calcein fluorescent dye or fluorescent intercalating dye [129,139]. The schematic procedure of RT-LAMP is shown as Figure 2.

Researchers have made efforts to optimize RT-LAMP for the development of rapid and sensitive detection of SARS-CoV-2. Several studies have evaluated the experimental parameters for RT-LAMP such as incubation temperature, incubation time, LoD, target genes, and primer sequences [143,144,145]. Aside from optimizing the experimental parameters, researchers have developed modified RT-LAMP procedures including methods without prior RNA extraction steps and high-throughput colorimetric assay methods using a 96-well plate format [146,147]. Modified RT-LAMP procedures also include methods coupled with Clustered Regularly Interspaced Short Palindromic Repeats (CRISPR) technology, a nanoparticle-based biosensor, and artificial intelligence [148,149,150,151].

### 3.2. Recombinase Polymerase Amplification (RPA)

Recombinase polymerase amplification is another isothermal amplification method that is widely used for SARS-CoV-2 detection. First described by Piepenburg et al. [152], RPA uses a complex of recombinase and two target specific primers (forward and reverse primers) to amplify the target nucleic acids [152]. Once the target nucleic acids are identified, recombinase-primer complex unwinds the target DNA and allows forward and reverse primers to hybridize [153]. The displaced DNA strand is amplified in the presence of DNA polymerase as primers elongate, and the template DNA is exponentially amplified until the reaction is completed [153]. RPA reaction is carried out at a single temperature between 37 and 42 °C, and the reaction is completed when ATPs are depleted, typically in less than an hour [154]. Amplified products are detected by gel electrophoresis, antigenic tags on primers and tag-specific antibodies, or fluorescent signals produced by a conjugated fluorophore and quencher on primers [152,153,154]. The schematic procedure of reverse transcription RPA (RT-RPA) is shown as Figure 2.

For SARS-CoV-2 detection, researchers have optimized RPA procedures by testing various experimental parameters that can now detect less than five viral copies from patient samples within 45 min from the sample collection [134]. RPA methods have also been optimized for SARS-CoV-2 detection by coupling RPA-based amplification with various CRISPR-based detection methods [155,156,157].

### 3.3. Other Isothermal Nuleic Acid Amplification Methods

Other than LAMP and RPA, isothermal amplification methods such as NASBA, SDA, and RCA have been used for the detection of SARS-CoV-2 [136,137,158]. Although we will not describe each technique in detail here, Table 3 presents the general features and components of each isothermal amplification method. 

## 4. Non-Conventional Methods

### 4.1. Genome Sequencing

Unprecedentedly massive genome sequencing has been undertaken with SARS-CoV-2 strains. The total number of sequenced genomes is approximately a million at present [30]. As the number of genomes increases rapidly, the World Health Organization (WHO) has provided guidelines for the genome sequencing of SARS-CoV-2 [159]. According to this guideline, the genome sequencing of SARS-CoV-2 can be used for understanding the emergence of SARS-CoV-2, understanding the biology of SARS-CoV-2, improving diagnostics and therapeutics, investigating virus transmission and spread, and inferring epidemiological parameters [159]. Based on the accumulative genome sequences of SARS-CoV-2, the emergence of the variants of concern [43,44,48,49,57,62], the origin of SARS-CoV-2 [2,31], and the mutation frequency of RT-qPCR primer/probe sites [30] can be known to humanity. Currently, various whole genome sequencing methods of the virus are being developed [160,161,162,163,164,165,166]. The sequencing methods of viruses can be categorized into the metagenomics approach and target enrichment based methods. In the metagenomics approach, the viral genome can be extracted from clinical samples and the extracted nucleic acid are sequenced. This can also be done with cultured viruses. These approaches have a clear advantage over target enrichment based methods. The metagenomics approach can be used even if there is no information of the pathogen or there are novel pathogens that were not previously known. However, a high proportion of host cell genetic materials can be found, which should be removed or reduced for sequencing. The removal or depletion methods of the host genetic materials vary by the type of sample or the virus [167,168,169,170,171,172,173,174]. Due to the nature of the metagenomics approaches, the clinical samples should ideally have a high titer of the pathogens. The metagenomics approach also can be done with cultured pathogens. However, the isolation and culturing of the pathogen are very time-consuming and labor-intensive work. In some cases, the isolation and culturing of some pathogens are not possible or are very difficult [175,176]. Alternatively, target enrichment methods can be used for the genome sequencing. The genetic materials of specific pathogens can be enriched through hybrid capture probes [177,178]. The sequences of the probes are complementary to the genome sequences of specific pathogens and these target enrichments effectively remove not-target sequences and increase the proportion of the target sequences. One of the advantages of hybrid capture approaches is the tolerance of sequence mismatch, allowing the capture of divergent variants. However, the hybrid capture approaches are relatively more expensive and complicated than other approaches. Another group of target specific enrichment approaches is the amplicon based approaches. The amplicon-based approaches are mainly dependent on PCR reactions. The PCR reaction can selectively enrich the genome of the target pathogens in the presence of non-target nucleic acids just like host genetic materials. Due to the nature of PCR, the amplicon based approaches are relatively more inexpensive, sensitive, and specific than other approaches. The WHO guidelines suggest that the complete genome sequencing can be done from the sample with Ct values of up to 30 and the partial genome sequencing can be done from the sample with Ct values of 30–35, although the Ct value can vary with various factors [159]. Currently, the most widely used primer panels for SARS-CoV-2 are ARTIC network amplicon sets [179]. At least three commercially available SARS-CoV-2 primer panels (CleanPlex SARS-CoV-2 Panel; Paragon genomics, QIAseq SARS-CoV-2 Primer Panel; Qiagen, and NEBNext ARTIC SARS-CoV-2 Library Prep Kit; NEB) are based on ARTIC network amplicon sets. However, there are also limitations. The design of primers requires prior knowledge of full sequence information of the genome. In addition, the primer intolerance to the sequence mismatch hinders the genome sequencing of the variants. The amplicon appears that the amplicon based approaches can be applied only to previously well-known pathogens. Alternatively, the genomic materials can be amplified sequence-independently [163,180,181]. Single primer isothermal amplification (SPIA) can amplify the genomic materials in a sequence-independent manner. As SPIA can amplify the genomic materials, prior knowledge of the pathogens is not required for a target enrichment approach. However, removing non-target genomic materials is mandatory for sequencing based SPIA as SPIA can also amplify non-target genomic material. Due to these characteristics, a high proportion of target genomic materials in the samples is crucial for successful SPIA based sequencing. SPIA based sequencing with low viral input showed very low coverage compared to other methods [180].

For the genome sequencing of pathogens, various sequencing technology can be used. While the conventional Sanger sequencing can still be used for viral genome sequencing [182], most SARS-CoV-2 genome sequencing is done with NGS sequencing technology. Currently, the most widely used NGS sequencing technology is the sequencing platforms of Illumina. Although the sequence length of individual reads is relatively short (paired-end 150 bp), the throughput and the accuracy of the individual reads are outstanding. Ion Torrent is another short reads sequencing platform technology, where the length of individual reads is 400 bp or 600 bp. The running time of the Ion Torrent sequencer is shorter than that of the Illumina sequencer. Long read alternatives are also available. The lengths of individual reads from PacBio and Oxford Nanopore Technology sequencers are tens of kilo base pairs or more. The individual reads from these long read sequences can cover most of the viral genome. However, the throughput of the long read sequencers is relatively lower than that of short read sequencers such as Illumina sequencing platforms. Furthermore, the accuracy of the individual reads is relatively lower than that of Illumina sequencing platforms. The sequencing platforms of Oxford Nanopore Technology maximize the benefits and drawbacks of the long read sequencer. The maximum length of the individual read is recorded up to a megabase scale [183]. However, due to the relative low accuracy of the individual reads, the WHO guidelines do not recommend these for SARS-CoV-2 genome sequencing unless the sequencing is replicated [159,184]. The coverage and depth of viral genome sequencing can be varied by the number of samples in single runs. Generally, most multiplex sequencing library kits for NGS support up to 384 samples per single runs. However, the production scale sequencers of Illumina (Hiseq, NextSeq, and NovaSeq) generate massive reads for small genome of viruses even with multiplex libraries. Though short individual reads require relatively more depth for high coverage, massive generation of sequence reads and low sequencing error rates of individual reads can compensate for short individual read length. Due to the massive sequence reads generation of the Illumina sequencer, the metagenomics approach of viral genome sequencing is practically available to only Illumina sequencer or similar platforms. Even if most of the sequence reads are non-target sequence reads (host genetic materials, contamination, etc.), a high quality genome assembly of the virus can be produced from the small remaining fraction of the target sequences. The long read sequencers such as sequencers by Pacbio and Oxford Nanopore Technology can generate very long individual reads that can cover most of the viral genome. However, due to the relatively low yield of total sequence reads and low accuracy of individual reads, the long read sequencers are not adequate for the metagenomics approach of viral genome sequencing. Instead, the long sequencers are more suitable for the amplicon-based approach. The target specific amplification can overcome the drawbacks of long read sequencers such as the low yield of total sequence reads and improving the accuracy of individual reads. Moreover, long sequencers can use long amplicons unlike short read sequencers. The schematic procedure of genome sequencing is shown in Figure 3.

### 4.2. CRISPR Based COVID-19 Detection

The clustered regularly interspaced short palindromic repeats (CRISPR)-Cas technique has been repurposed for diagnostics and is one of the widely used nucleic acid detection methods [185,186]. Many types of Cas proteins have been developed to create highly accurate and sensitive diagnostic methods [187]. Cas9 has been widely used for genome editing while DNA-targeting Cas12 (also known as Cpf1 or C2c1) effectors and RNA-targeting Cas 13a are more suitable for disease diagnosis [188]. 

Compared to conventional diagnostic methods such as RT-qPCR, CRISPR-based approaches can quickly provide rapid, visual, highly sensitive, and specific detection due to the collateral cleavage of a reporter dye in the presence of a target [189].

Numerous CRISPR-Cas detection systems have been developed. For example, techniques of a CRISPR–Cas12-based assay have been developed for the detection of SARS-CoV-2 from patient sample RNA, called SARS-CoV-2 DNA Endonuclease-Targeted CRISPR Trans Reporter (DETECTR) [190]. This assay includes simultaneous reverse transcription and isothermal amplification using loop-mediated amplification (RT–LAMP) [191]. An RNA targeting Cas13a dependent platform [156], the SHERLOCK (Specific High Sensitivity Enzymatic Reporter UnLOCKing) technique, offers a simplified test and has a limit of detection of 100 copies of the viral genome [192,193].

Recently, a lyophilized CRISPR-Cas12 assay for SARS-CoV-2 detection (Lyo-CRISPR SARS-CoV-2 kit) has been developed based on reverse transcription (RT), isothermal amplification, and CRISPR-Cas12 reaction [194]. The schematic procedure of CRISPR detection systems is shown in Figure 1D.

### 4.3. Nanotechnology Based Methods

Nanotechnology has already proven its value through its diagnostic, vaccine, and therapeutic applications that have expanded into clinical applications [195]. Scientists have shown that nucleic acid detection using nanomaterials for viral infectious diseases now have various advantages in the diagnostic field [196]. Moreover, nanomaterials are powerful tools for the diagnosis, prevention, and treatment of COVID-19 [197].

Magnetic nanoparticles have been used in RT-qPCR diagnosis for the extraction of viral RNA from SARS-CoV-2. This method merges the lysis and binding steps, and the poly (amino ester) with carboxyl groups (PC)-coated magnetic nanoparticles (pcMNPs)-RNA complexes can be directly introduced into RT-qPCR reactions [198]. In addition, a test has been developed to diagnose SARS-CoV-2 that can rapidly detect the virus. The test is performed using gold nanoparticles to detect specific proteins such as nucleocapsid phosphoprotein. In the presence of gold nanoparticles, the test is positive upon the color of the liquid reagent changing from purple to blue [199]. Nanoparticles that can be applied in the RT-qPCR diagnosis are shown in Figure 1E.

## 5. Resources and Standardization for SARS-CoV-2 Nucleic Acid Testing

### 5.1. International Activity for Standardization

The Consultative Committee on the Quantity of Material (CCQM) is responsible for developing and documenting the equivalence of national standards such as certified reference materials (CRMs)/reference materials (RMs) and reference methods for biological and chemical measurements [200]. It advises the International Committee for Weights and Measures (Comité international des poids et mesures, CIPM) on matters related to biological and chemical measurements including advice on the International Bureau of Weights and Measures’ (Bureau international des poids et mesures, BIPM) scientific program activities [200]. One of the responsibilities of the CCQM is to contribute to the development of a worldwide admitted system of national measurement standards, reference methods, and facilities for biological and chemical measurements [201]. Several National Metrology Institutes (NMIs) and expert laboratories from many countries have performed highly sensitive and accurate measurements of the amount of the SARS-CoV-2 viral RNA tested using reverse transcription-digital PCR (RT-dPCR) [202]. The capability to accurately measure the amount of COVID-19 causing viral nucleic acid with equivalence globally will remarkably improve diagnostic testing confidence and support countries in effectually tackling the pandemic situation [202]. Moreover, the biological, chemical, and physical measurement ability of NMIs are enabling industry and laboratories to effectively and quickly face the COVID-19 challenge [203].

NMIs support quality assurance by developing and providing CRM/RM for COVID-19 and ensures equivalent pathology testing, while minimizing false negative and positive test outcomes.

Non-NMIs including both public and private sectors have developed standards and reference materials for SARS-CoV-2 RNA. The National Institute for Biological Standards and Control (NIBSC) distributed the 1st WHO International Standards for SARS-CoV-2 RNA in 2020, which is used for the standardization of nucleic acid amplification technique (NAT)-based diagnostic assays. The U.S. Food and Drug Administration developed the SARS-CoV-2 Reference Panel in 2020 to precisely compare the performance of NAT-based assays for SARS-CoV-2 detection. Non-profit organizations such as American Type Culture Collection (ATCC) and private industries such as LGC SeraCare have also developed and distributed reference materials for NAT-based SARS-CoV-2 diagnosis.

### 5.2. Reference Materials

A RM can be defined as a “material, sufficiently homogeneous and stable with respect to one or more specified properties, which has been established to be fit for its intended use in a measurement process” [204]. According to the general requirements for the competence of reference material producers (ISO 17034:2016), a CRM is defined as a “reference material characterized by a metrologically valid procedure for one or more specified properties, accompanied by a reference material certificate that provides the value of the specified property, its associated uncertainty, and a statement of metrological traceability” [204]. Generally, major producers of RMs are NMIs [205]. However, RMs of SARS-CoV-2 are produced by not only NMIs, but also by commercial institutes and culture collections. The first RM of SARS-CoV-2 was produced by the National Institute of Metrology of China (NIMC). The RM was synthetic RNA based on the genome sequence of Wuhan-Hu-1, the first identified SARS-CoV-2 strain. The RM of NIMC was quantified with the ddPCR method and contained N, E, and partial genes of RdRp genes that can cover the WHO announced in-house assay for SARS-CoV-2 [91]. Other NMIs such as the Joint Research Center in Europe (JRC), Korea Research Institute of Standards and Science (KRISS), National Measurement Institute of Australia (NMIA), National Institute of Standards and Technology (NIST), and The National Metrology Institute of Turkey (UME) have also produced RMs for SARS-CoV-2 nucleic acid testing. NIBSC also produced a RM for SARS-CoV-2 nucleic acid testing and it has been used as WHO reference materials. Type culture collection also produced RMs. ATCC and BEI resources produced RMs with inactivated SARS-CoV-2 strain (USA-WA1/2020). Not only public institutes but also commercial institutes have produced RMs. Seracare, which is a subsidiary of LGC, produced RMs in the early phase of the pandemic. The RMs were SARS-CoV-2 RNA covered with viral proteins, which can be used for the WHO announced in-house assays. Twist Bioscience produced RMs with various SARS-CoV-2 strains including emerging variants such B.1.1.7, B.1.351, and P.1.

The majority of these RMs were quantitated with dPCR. The comparison of qPCR and dPCR methods showed that dPCR can quantitate specific genes regardless of primer sequences [72], indicating that the dPCR assay is a suitable method for the measurement of RMs. The reference materials of SARS-CoV-2 nucleic acid testing are listed in Table 4 and Appendix A.

### 5.3. Other Resources

The genomic sequences of SARS-CoV-2 are mainly deposited in GISAID databases and are also available at the NCBI GenBank [27,30]. The metadata of strains such as collection date, patient age, gender, sequencing methods, etc. are also available in GISAID. There are also other specialized databases. Virus Pathogen Resources (ViPR) provides detailed genome sequence information and analysis tools [206]. Nextstrain provides real-time tracking evolution and spreading of SARS-CoV-2 [28]. Nextstrain visualizes tracking of SARS-CoV-2 lineages by integrating geographic information and sequence information. CoV-GLUE is a mutation dedicated database for SARS-CoV-2 [207]. CoV-GLUE summarizes the mutations of amino acid replacements, insertions, and deletion.

Although more than a million SARS-CoV-2 genome sequences are publicly available, live SARS-CoV-2 sources are relatively limited. As SARS-CoV-2 is regarded as risk group 3 and handling of live SARS-CoV-2 requires biosafety level 3 laboratories, only a few culture collections distribute SARS-CoV-2 strains. BEI resources have been established by the National Institute of Allergy and Infectious Diseases (NIAID) and provide live SARS-CoV-2 strains and derivatives of SARS-CoV-2. The National Culture Collection for Pathogens of Korea (NCCP) also provides live viruses and derivatives from SARS-CoV-2 isolated in Korea. European Virus-Archive Global (EVAg) is a network between 25 laboratories including 16 EU member state institutions and nine non-EU institutions. It also provides isolated strains and derivatives from the EU and related countries. Training courses such as SARS-CoV-2 diagnostic training are also available at EVAg.

## 6. Conclusions

Currently, representative diagnostic methods of SARS-CoV-2 are RT-qPCR assays. Most countries use the RT-qPCR assay as a primary method for diagnostics. Though alternative methods are available, their sensitivity, specificity, or costs are not comparable to RT-qPCR assays. However, RT-qPCR assays require relatively expensive instruments and highly trained personnel. These requirements restrict the expansion of diagnostics capacity in some countries. To overcome these drawbacks, future diagnostics methods should be inexpensive and simple, which can be used for point-of-care testing. For example, if inexpensive whole viral genome sequencing methods are developed in the future, they will be capable of replacing RT-qPCR assays as a standard disease diagnostic test. The whole genome sequencing of the virus can provide more information including variant information of the viruses. The reference materials for the diagnostics are also important to assess newly developed diagnostic methods.

## Figures and Tables

**Figure 1 ijms-22-06150-f001:**
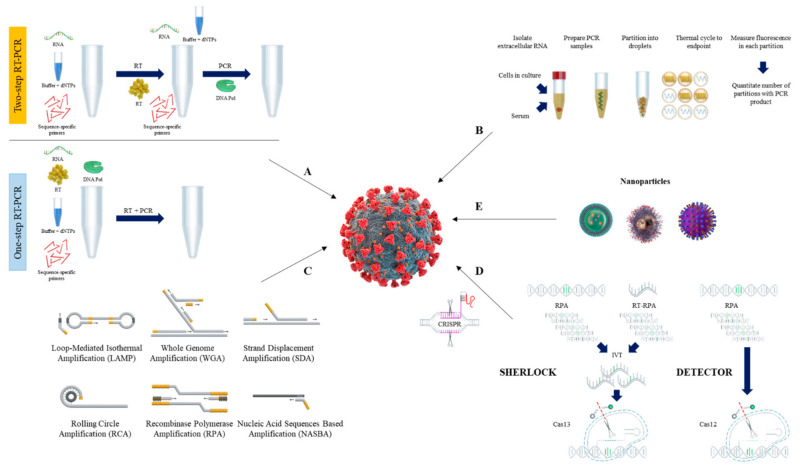
Overview of nucleic acid testing for SARS-CoV-2. The schematic procedure of RT-qPCR (**A**), and dPCR (**B**). Current isothermal amplification methods (**C**), CRISPR detection systems (**D**), and nanoparticles (**E**) are also shown.

**Figure 2 ijms-22-06150-f002:**
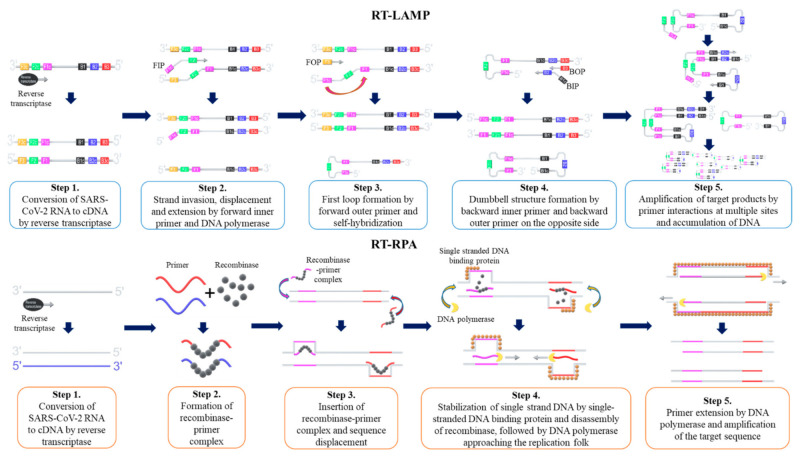
Schematic procedure of reverse transcription loop-mediated isothermal amplification (RT-LAMP) and reverse transcription recombinase polymerase amplification (RT-RPA). FIP = Forward Inner Primer, FOP = Forward Outer Primer, BIP = Backward Inner Primer, BOP = Backward Outer Primer.

**Figure 3 ijms-22-06150-f003:**
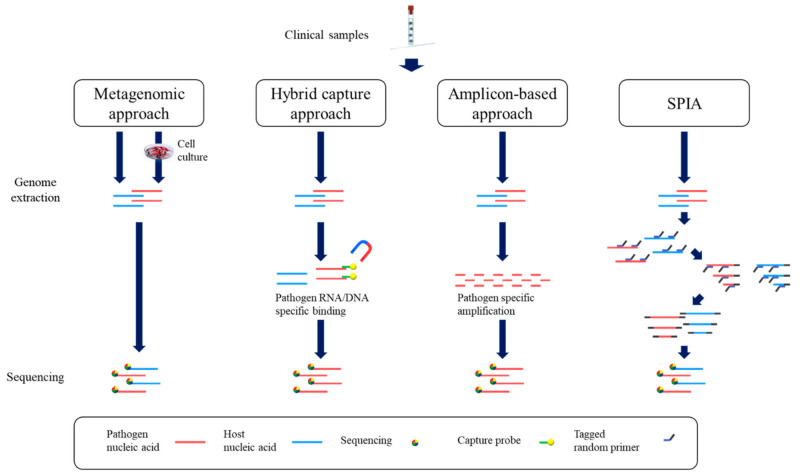
Schematic procedure of genome sequencing.

**Table 1 ijms-22-06150-t001:** Nomenclature of SARS-CoV-2.

GISAID Clades	PANGO Lineage	Nextstrain Clades	Notable Variants
S	A	19B	A.23.1
L	B	19A	Wuhan-Hu-1
V			
G	B.1	20A	B.1.525, B.1.627
GH	B.1	20C	B.1.427, B.1.429, B.1.526
	B.1.2	20G	
	B.1.596		
	B.1.351	20H/501.Y.V2	B.1.351
GR	B.1.1.1	20B	
	P.3		P.3
	C	20D	
	D.2	20F	
	P.1	20J/501.Y.V3	P.1
GV	B.1.177	20E (EU1)	B.1.177
GRY	B.1.1.7	20I/501.Y.V1	B.1.1.7

**Table 2 ijms-22-06150-t002:** Notable variants of SARS-CoV-2.

PANGO Lineage	CDC Designation	PHE Designation	First Detected	Spike Protein Substitutions
B.1.1.7	VOC	VOC-20DEC-01, VOC-21FEB-02 *	United Kingdom	69del, 70del, 144del, (E484K), (S494P), N501Y, A570D, D614G, P681H, T716I, S982A, D1118H (K1191N)
B.1.351	VOC	VOC-20DEC-02	South Africa	D80A, D215G, 241del, 242del, 243del, K417N, E484K, N501Y, D614G, A701V
P.2	VOI	VUI-21JAN-01	Brazil	E484K, (F565L), D614G, V1176F
P.1	VOC	VOC-21JAN-02	Brazil	L18F, T20N, P26S, D138Y, R190S, K417T, E484K, N501Y, D614G, H655Y, T1027I
A.23.1	-	VUI-21FEB-01 *	Uganda	F157L, V367F, (E484K), Q613H, P681R
B.1.525	VOI	VUI-21FEB-03	United Kingdom	A67V, 69del, 70del, 144del, E484K, D614G, Q677H, F888L
B.1.1.318	-	VUI-21FEB-04	United Kingdom	D614G, D796H, E484K, P681H, T95I, 144del
P.3	-	VUI-21MAR-02	Philippines	E484K, N501Y, P681H
B.1.617	VOI	VUI-21APR-01	India	L452R, E484Q, D614G
B.1.617.2	VOI	VOC-21APR-02	India	T19R, (G142D), 156del, 157del, R158G, L452R, T478K, D614G, P681R, D950N
B.1.617.3	VOI	VUI-21APR-03	India	T19R, G142D, L452R, E484Q, D614G, P681R, D950N
AV.1	-	VUI-21MAY-01	United Kingdom	D80G, T95I, G142D, 144del, N439K, E484K, D614G, P681H, I1130V, D1139H
B.1.617.1	VOI	-	India	(T95I), G142D, E154K, L452R, E484Q, D614G, P681R, Q1071H
B.1.526	VOI	-	United States	(L5F), T95I, D253G, (S477N), (E484K), D614G, (A701V)
B.1.526.1	VOI	-	United States	D80G, 144del, F157S, L452R, D614G, (T791I), (T859N), D950H
B.1.427	VOC	-	United States	L452R, D614G
B.1.429	VOC	-	United States	S13I, W152C, L452R, D614G

* with E484K; ( ) detected in some sequences but not all; VOC; Variant of Concern, VOI; Variant of Interest, VUI; Variant under Investigation.

**Table 3 ijms-22-06150-t003:** General features of the isothermal amplification techniques for SARS-CoV-2 detection.

Method	Components	Temperature	Time	Detection Method	Advantages *	Disadvantages *
Loop-mediated isothermal amplification (LAMP)	DNA polymerase, forward inner primer, backward inner primer, forward outer primer, backward outer primer	60–65 °C	>1 h	Colorimetric, turbimetric, fluorescence probe, intercalating dye	High specificity. Less sensitive to inhibitors in biological samples	False positive in negative control
Recombinase polymerase amplification (RPA)	Recombinase, single stranded binding protein, DNA polymerase, forward primer, reverse primer	37–42 °C	>1 h	Fluorescence, antigenic-tag (antibody)	Performed in the presence of PCR inhibitors. Fast and sensitive	Inhibited by detergents (SDS and CTAB). Non-specific/high background signal
Nucleic acid sequence-based amplification (NASBA)	RNase H, reverse transcriptase, T7 DNA-dependent RNA polymerase, forward primer with T7 promoter sequence, reverse primer	41 °C	>2 h	Fluorescence	More sensitive and less time-consuming	Non-specific reactions/false positives
Strand-displacement amplification (SDA)	DNA polymerase, restriction endonuclease, primers, dCTP, dTTP, dGTP, dATPα	37–49 °C	>2 h	Fluorescence	High specificity. Detection of large RNA molecules	Non-specific reaction/high background signal
Rolling circle amplification (RCA)	DNA ligase, DNA polymerase, primer, padlock probe	30–37 °C	>1.5 h	Fluorescence	High specificity	False negatives and false positives

* Advantages and disadvantages in comparison with RT-qPCR methods.

**Table 4 ijms-22-06150-t004:** Reference materials and resources for SARS-CoV-2 nucleic acid testing.

Institute	Type	Numbers
ATCC	heat-inactivated	3
	Synthetic RNA	5
Bio-Rad	Synthetic RNA	1
JRC	Synthetic RNA	1
KRISS	Synthetic RNA	1
	Virus Like Particle	1
NIBSC	Heat-inactivated	1
NMIA	Inactivated	1
NIMC	Synthetic RNA	1
NIST	Synthetic RNA	1
Randox Qnostics	Heat-inactivated	3
Seracare	Virus Like Particle	4
Thermo Scientific	Inactivated	1
	Genomic RNA	1
	Synthetic RNA	2
Twist Bioscience	Synthetic RNA	20
UME	Synthetic RNA	2
ZeptoMetrix	Chemical-inactivated	1

ATCC; American Type Culture Collection, JRC; Joint Research Center, KRISS; Korea Research Institute of Standards and Science, NIBSC; National Institute for Biological Standards and Control, NMIA; National Measurement Institute Australia, NIMC; National Institute of Metrology of China, NIST; National Institute of Standards and Technology, UME; The National Metrology Institute of Turkey.

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
