# Peer review of "Nucleic Acid Testing of SARS-CoV-2"

_ijms, 2021, doi:10.3390/ijms22116150_

Round 1

Reviewer 1 Report

Dear authors,

I find the presented manuscript an interesting review of molecular methods. What I find as an extra value is addressing reference materials. There are some minor text revisions that you should correct (Sentence line 38-40 not clear). In my opinion, this review should have a paragraph with conclusions and a synthesis of major issues for the molecular diagnosis of SARS-CoV-2 and possible solutions. The availability of reference materials and methods that are up to date with new viral variants is one issue that should be addressed.

Author Response

Response to Reviewer 1

I find the presented manuscript an interesting review of molecular methods. What I find as an extra value is addressing reference materials.

  • Answer to reviewer

We are appreciated for your thoughtful comments on our manuscript.

  1. There are some minor text revisions that you should correct (Sentence line 38-40 not clear).
  • Answer to reviewer

We revised the sentence.

  1. In my opinion, this review should have a paragraph with conclusions and a synthesis of major issues for the molecular diagnosis of SARS-CoV-2 and possible solutions.
  • Answer to reviewer

The conclusion section was added.

  1. The availability of reference materials and methods that are up to date with new viral variants is one issue that should be addressed.
  • Answer to reviewer

The table 2, 4, and S1 were updated.

Reviewer 2 Report

The review paper on “Nucleic Acid Testing of SARS-CoV-2” presents an exhaustive summary of the importance of an accurate and rapid diagnostic routine and standardized methods for the Covid-19 pandemic. The authors coved the use of RT-PCR, digital PCR, LAMP isothermal, and RT- 19 RPA and non-classical methods such as NGS and CRISPR-based assays.

Comments:

  1. Yoo et al covers the RNA/DNA diagnosis methods but often ‘slip’ to side-walk issues. For example. “Even now, the number of new cases continues…. mechanisms of SARS-CoV-2 infection” it continues about development new therapies and claim to cover basic biology of the virus, origin, pathophysiology, and diagnosis. Another example line 295-310 are some discussion on metagenomics.  

I suggest removing all these claims and restrict the focus on ‘diagnosis by genetic materials’ as the title indicates.

  1. A summary of what seems to be the direction of the SARS-CoV-2 detection methods and the challenges ahead are missing (e.g., benefit for quick serological tests)  
  2. Add a section of (main) abbreviations as it is easy to get lost.
  3. Table 2. Add for each listed variant the number of missense mutations versus the original Whuan version (also deletion/ insertions).
  4. Important (practical) consideration on RT-PCR (page 5) should be mentioned explicitly (as this is the ‘gold standard’ as the authors correctly claim). Examples for such missing information include:
    1. Sampling issues – RT-PCR is done in parallel on human cells (to confirm sampling quality)
    2. Definition of the degree of ct values (for RT-PCR) as SArS-CoV-2 positive, maybe or negatives. Also, Ct value itself cannot be directly interpreted as viral load without a standard curve 10.1016/S1473-3099(20)30424-2.
    3. Minimal number of viruses in the samples for detection
    4. False-negative/ False positive. In case information is available to different technologies it is worth mentioning (e.g. 10.1101/2020.09.25.20201921)
    5. Sample source and its success (saliva, nasal swab)
    6. The difference in sensitivity for each of the tested gene (e.g. Eurosurveillance 25.21 (2020): 2000880).
    7. Pooling technologies of RT-PCR (for saving time, money, screening population as in schools).
  5. Figure 1 is confusing. There are 6-7 methods mentioned graphically. Identified them by A, B, C and correctly refer to them in the text. Not clear what is the motivation to add them graphically of liposomes, VLNP?, inorganic Gold? etc. For most readers, those are abbreviations with no meetings. Make it simpler. RT-PCR (top left) is a complicated scheme that complicates matter.
  6. Figure 2 will benefit from a legend to the many graphical ‘symbols’ (as in Fig 3). The use of small unreadable fonts should be avoided.
  7. Some pages are uneasy to read (e.g. According to ISO 17034:2016, a CRM is defined as a “reference material…” and more)
  8. Table 4 is annoying to read. Provide a dense and easy-to-read sampling for example all the Twist Bioscience (1-17) can be compressed to one line. All the 17 detailed control can be moved to supplemental.
  9. Some professional discussion on NGS methodologies remains unclear how it is connected to the SARS-CoV-2 practicality (e.g., the section on nanopore, PacBio vs Illumina NGS (e.g., length, depth, mistake and accuracy). Lines 340-360.
  10. Section 4.3 is mostly on the importance of nanotechnology in medicine. I believe it is another deviation from the main topic and does not contribute much to the main topic.

Small edits:

  1. Table 1 – PANGo correlt to PANGO
  2. Change ACEII to ACE2
  3. Page 5, line 171 rephrase
  4. 1 simplified and mark A, B, C
  5. Table 4. Needs revision. Header 692314 ? Replace to simple terms: “Genomic RNA from Severe acute respiratory syndrome related coronavirus 2” is “RNA, CoV-2 related”. SARS-Related Coronavirus 2 (SARS-CoV-2) Negative Control, simplify. “RNAAssay Ready Control” replace by Ready Control. Some unexplained abbreviation (NAT?)

Author Response

Response to Reviewer 2

The review paper on “Nucleic Acid Testing of SARS-CoV-2” presents an exhaustive summary of the importance of an accurate and rapid diagnostic routine and standardized methods for the Covid-19 pandemic. The authors coved the use of RT-PCR, digital PCR, LAMP isothermal, and RT- 19 RPA and non-classical methods such as NGS and CRISPR-based assays.

  • Answer to reviewer

We are appreciated for your constructive comments on our manuscript. Your comments are very helpful to improve our manuscript.

  1. Yoo et al covers the RNA/DNA diagnosis methods but often ‘slip’ to side-walk issues. For example. “Even now, the number of new cases continues…. mechanisms of SARS-CoV-2 infection” it continues about development new therapies and claim to cover basic biology of the virus, origin, pathophysiology, and diagnosis. Another example line 295-310 are some discussion on metagenomics.

I suggest removing all these claims and restrict the focus on ‘diagnosis by genetic materials’ as the title indicates.

  • Answer to reviewer

Some of these parts are introduction for the related topics. For example, the line 295-310 were introductory parts for practically used or commercially available for SARS-CoV-2 sequencing.

  1. A summary of what seems to be the direction of the SARS-CoV-2 detection methods and the challenges ahead are missing (e.g., benefit for quick serological tests)
  • Answer to reviewer

The conclusion section was added and the topics was described in the section.

  1. Add a section of (main) abbreviations as it is easy to get lost.
  • Answer to reviewer

The abbreviations section was added.

  1. Table 2. Add for each listed variant the number of missense mutations versus the original Whuan version (also deletion/ insertions).
  • Answer to reviewer

Spike protein substitution was added to table 2.

  1. Important (practical) consideration on RT-PCR (page 5) should be mentioned explicitly (as this is the ‘gold standard’ as the authors correctly claim). Examples for such missing information include:
  2. Sampling issues – RT-PCR is done in parallel on human cells (to confirm sampling quality)
  3. Definition of the degree of ct values (for RT-PCR) as SArS-CoV-2 positive, maybe or negatives. Also, Ct value itself cannot be directly interpreted as viral load without a standard curve 10.1016/S1473-3099(20)30424-2.
  4. Minimal number of viruses in the samples for detection
  5. False-negative/ False positive. In case information is available to different technologies it is worth mentioning (e.g. 10.1101/2020.09.25.20201921)
  6. Sample source and its success (saliva, nasal swab)
  7. The difference in sensitivity for each of the tested gene (e.g. Eurosurveillance 25.21 (2020): 2000880).
  8. Pooling technologies of RT-PCR (for saving time, money, screening population as in schools).
  • Answer to reviewer

We added important considerations on RT-PCR with the missing information that you suggested.

  1. Figure 1 is confusing. There are 6-7 methods mentioned graphically. Identified them by A, B, C and correctly refer to them in the text. Not clear what is the motivation to add them graphically of liposomes, VLNP?, inorganic Gold? etc. For most readers, those are abbreviations with no meetings. Make it simpler. RT-PCR (top left) is a complicated scheme that complicates matter.
  • Answer to reviewer

We added A, B, C… near the arrows and correctly referenced them in the manuscript. We also removed most of the abbreviations and graphics for nanoparticles for simplicity. Moreover, we replaced the scheme of RT-PCR with a simpler diagram

  1. Figure 2 will benefit from a legend to the many graphical ‘symbols’ (as in Fig 3). The use of small unreadable fonts should be avoided
  • Answer to reviewer

We directly labelled each graphical component (symbol) on the figure. We also used bigger fonts for each label

  1. Some pages are uneasy to read (e.g. According to ISO 17034:2016, a CRM is defined as a “reference material…” and more)
  • Answer to reviewer

The sentences were revised.

  1. Table 4 is annoying to read. Provide a dense and easy-to-read sampling for example all the Twist Bioscience (1-17) can be compressed to one line. All the 17 detailed control can be moved to supplemental.
  • Answer to reviewer

The table 4 was replaced with a concise one. The original table was move supplementary table

  1. Some professional discussion on NGS methodologies remains unclear how it is connected to the SARS-CoV-2 practicality (e.g., the section on nanopore, PacBio vs Illumina NGS (e.g., length, depth, mistake and accuracy). Lines 340-360.
  • Answer to reviewer

The discussion was added the end of genome sequencing section.

  1. Section 4.3 is mostly on the importance of nanotechnology in medicine. I believe it is another deviation from the main topic and does not contribute much to the main topic.
  • Answer to reviewer

We deleted the sentences regarding the importance of nanotechnology in medicine as it is not related to the main topic.

Small edits:

  1. Table 1 – PANGo correlt to PANGO
  2. Change ACEII to ACE2
  3. Page 5, line 171 rephrase

*Changes: Line X-X

  1. 1 simplified and mark A, B, C

*Changes: Figure 1

  1. Table 4. Needs revision. Header 692314 ? Replace to simple terms: “Genomic RNA from Severe acute respiratory syndrome related coronavirus 2” is “RNA, CoV-2 related”. SARS-Related Coronavirus 2 (SARS-CoV-2) Negative Control, simplify. “RNAAssay Ready Control” replace by Ready Control. Some unexplained abbreviation (NAT?)
  • Answer to reviewer

The sentence and table were revised as reviewer’s comments.

Round 2

Reviewer 1 Report

Dear Authors,

Thank you for addressing my concerns. I suggest that article should be accepted in present form

Kind regards

Reviewer 2 Report

few minor (no need for another referee cycle) items

  1. Fig 1 the arrows of A-E should be reversed (from the virus to the method and not the opposite way)
  2. Ref 84-85 are duplicated. Keep only 84
  3. few formatting issues